# Tetrodotoxin/Saxitoxin Accumulation Profile in the Euryhaline Marine Pufferfish *Chelonodontops patoca*

**DOI:** 10.3390/toxins16010018

**Published:** 2023-12-28

**Authors:** Hongchen Zhu, Takashi Sakai, Hiroyuki Doi, Kenichi Yamaguchi, Akinori Yamada, Tomohiro Takatani, Osamu Arakawa

**Affiliations:** 1Graduate School of Fisheries and Environmental Sciences, Nagasaki University, 1-14, Bunkyo-machi, Nagasaki 852-8521, Japan; zhc957286316@hotmail.com (H.Z.); sakatakashi1118@docomo.ne.jp (T.S.); kenichi@nagasaki-u.ac.jp (K.Y.); ayamada@nagasaki-u.ac.jp (A.Y.); taka@nagasaki-u.ac.jp (T.T.); 2Nifrel, Osaka Aquarium Kaiyukan, 2-1, Senribanpakukoen, Suita, Osaka 565-0826, Japan; doi@kaiyukan.com

**Keywords:** *Chelonodontops patoca*, pufferfish, tetrodotoxin (TTX), saxitoxin (STX), intrarectal administration

## Abstract

Marine *Takifugu* pufferfish, which naturally possess tetrodotoxins (TTXs), selectively take up and accumulate TTXs, whereas freshwater *Pao* pufferfish, which naturally possess saxitoxins (STXs), selectively take up and accumulate STXs. To further clarify the TTXs/STXs selectivity in pufferfish, we conducted a TTX/STX administration experiment using *Chelonodontops patoca*, a euryhaline marine pufferfish possessing both TTXs and STXs. Forty nontoxic cultured individuals of *C. patoca* were divided into a seawater group (SW, acclimated/reared at 33‰ salinity; *n* = 20) and a brackish water group (BW, acclimated/reared at 8‰ salinity; *n* = 20). An aqueous TTX/STX mixture was intrarectally administered (both at 7.5 nmol/fish), and five individuals/group were analyzed after 1–48 h. Instrumental toxin analyses revealed that both TTX and STX were taken up, transferred, and retained, but more STX than TTX was retained in both groups. TTX gradually decreased and eventually became almost undetectable in the intestinal tissue, while STX was retained at ~5–10% of the dose level, and only STX showed transient transfer in the liver. The BW group showed a faster decrease/disappearance of TTX, greater STX retention in the intestine, and greater STX transient transfer to the liver. Thus, *C. patoca* appears to more easily accumulate STXs than TTXs, especially under hypoosmotic conditions.

## 1. Introduction

Tetrodotoxins (TTXs), a group of highly potent neurotoxins consisting of TTX and its analogues, are commonly detected in a variety of organisms, including pufferfish, oysters, newts, tropical gobies, frogs, blue-ringed octopuses, marine snails, starfish, flatworms, ribbon worms, xanthid crabs, and horseshoe crabs [1,2,3,4,5]. *Takifugu* is a marine pufferfish genus found in the coastal areas of Japan that bears TTXs as its principal toxin component [1,2,6,7,8,9]. Although the precise origin of TTXs in pufferfish remains unclear, it is speculated that TTX is not biosynthesized by pufferfish, but is rather accumulated through the food chain originating from TTX-producing bacteria [1,10,11,12]. Pufferfish that are separated from the natural food chain and reared on nontoxic food from hatching are nontoxic [13]. In vivo TTX administration studies of individuals of such nontoxic cultured *Takifugu* have revealed the unique TTX kinetics of marine pufferfish; TTX is taken up into the blood from the intestine, retained in the liver, and then gradually transferred to the skin/ovaries [13,14]. These unique kinetics are thought to involve isoforms of the toxin-binding protein PSTBP (pufferfish saxitoxin [STX] and TTX binding protein) present in pufferfish plasma [15]. Whereas TTXs are primarily distributed in the liver and ovaries as well as in the skin of some species of the marine *Takifugu* adult pufferfish, the testes and muscle are not toxic or weakly toxic [1,16,17]. The toxin distribution in the body may reflect the TTX kinetics.

Saxitoxins (STXs), including STX, gonyautoxins (GTXs), and C toxins, are also potent neurotoxins belonging to a group of neurotoxic alkaloids that were frequently associated with toxifying filter-feeding invertebrates, including bivalves, crustaceans, and ascidians [18,19,20,21]. The molecular weight, toxicity, and intoxication mechanism of the main components are similar between STX and TTX [22,23,24]. *Pao* and *Leiodon* genera, both previously known as *Tetraodon*, are freshwater pufferfish which generally live in the Mekong Basin of Southeast Asian countries and in the rivers and lakes of South Asian countries such as Bangladesh, and have no TTXs but contain STXs as the only toxic component [13,25,26]. STXs in pufferfish are also exogenous and are presumed to accumulate through the food chain beginning with marine dinoflagellates that produce STXs and STXs-producing freshwater cyanobacteria [20,27,28,29,30,31]. These freshwater pufferfish possess high concentrations of STXs in their ovaries and skin. Ngy et al. administered decarbamoylsaxitoxin (dcSTX) intramuscularly to nontoxic cultured individuals of the freshwater pufferfish *Pao turgidus* and found that dcSTX was selectively transferred via the blood from the muscle to the skin, where the majority of toxin accumulated, suggesting that freshwater pufferfish also have unique internal STXs kinetics, in which the skin is a crucial site of accumulation [13].

Some genera or species of marine pufferfish possess both TTXs and STXs, such as the genus *Sphoeroides* in Florida, *Arothron* in the Philippines and Japanese coastal waters, and *Canthigaster* in the Caribbean Sea [13,27,28,30]. The TTXs/STXs ratio in these pufferfish varies among species and tissues, but STXs are the major component in some cases. Recently, Zhu et al. [32] examined the toxin profile and distribution in the body of the marine pufferfish *Canthigaster valentini* from the coastal waters of Japan and showed that the skin and ovaries accumulate a large amount of toxin; the major toxic component is TTX and approximately 20% is STXs. On the whole, marine pufferfish of the genus *Takifugu* are thought to primarily possess TTXs; freshwater pufferfish of the genera *Pao* and *Leiodon* possess only STXs; and marine pufferfish such as *Sphoeroides*, *Arothron*, and *Canthigaster* could possess both TTXs and STXs simultaneously. To clarify the origin of these differences in toxin profiles, we recently conducted oral gavage TTX/STXs administration experiments using nontoxic cultured individuals of the marine pufferfish *Takifugu pardalis*, which naturally possesses TTXs, and the freshwater pufferfish *Pao suvattii*, which naturally possesses STXs. Our findings revealed that *T. pardalis* and *P. suvattii* selectively take up and accumulate TTX and STXs, respectively [33]. In addition, tissue slice incubation experiments using the liver and other tissues showed that when TTX/STXs are present, the tissues of *Takifugu rubripes* selectively take up TTX [34,35,36] and tissues of the *Dichotomyctere fluviatilis* (a euryhaline freshwater pufferfish) selectively take up STXs [37]. Thus, the ratio of TTXs/STXs in a pufferfish is dependent not only on the abundance of TTXs/STXs in its habitat (external factor), but also on its inherent TTXs/STXs selectivity (internal factor). TTXs/STXs selectivity at the level of the individual and/or tissue in species that possess TTXs and STXs simultaneously is unclear.

*Chelonodontops patoca* (formerly known as *Chelonodon patoca*) is a euryhaline marine pufferfish inhabiting coastal waters, including reef-associated areas and tropical waters [38]. These pufferfish, which inhabit the subtropical waters of Japan, such as the coastal waters of Okinawa Prefecture and Amami-Oshima Island, Kagoshima Prefecture, possess large amounts of TTXs in their skin and ovary, but there have been no reports of STXs being detected [39,40]. On the other hand, in addition to TTXs, STXs equivalent to approximately 16% of the total toxin have been detected in individuals collected in tropical waters off the coast of the Philippines [27]. Therefore, *C. patoca*, like pufferfish of the genera *Arothron* and *Canthigaster*, are thought to be capable of possessing TTXs and STXs simultaneously, although they have a greater preference for TTXs. In this study, to test our hypothesis mentioned above that the ratio of TTXs/STXs in a pufferfish is dependent on its inherent TTXs/STXs selectivity, we conducted an intrarectal TTX/STX administration experiment using nontoxic cultured individuals of a species that possesses TTXs and STXs simultaneously, namely *C. patoca* (Figure 1).

Marine pufferfish primarily have TTXs, whereas freshwater pufferfish possess only STXs. One of the most significant differences between marine and freshwater ecosystems is salinity (osmotic pressure), and such external factors may play a crucial role in TTXs/STXs selectivity in pufferfish. Therefore, under the hypothesis that STXs selectivity increases (or TTXs selectivity decreases) in low-salinity conditions, we also examined the effect of salinity on the toxin uptake and accumulation profiles.

## 2. Results

### 2.1. TTX/STX Accumulation Profile of C. patoca

Forty cultured *C. patoca* individuals were divided into two groups of 20 individuals each; a seawater group (SW, acclimated/reared at 33‰ salinity) and a brackish water group (BW, acclimated/reared at 8‰ salinity). The fish in the two groups were intrarectally administered an aqueous mixture of TTX and STX (both at 7.5 nmol/fish), and five individuals in each group were analyzed after 1, 12, 24, and 48 h, respectively. Instrumental analyses of the toxin content in the intestine, muscle, liver, and skin (typical chromatograms are shown in Figure A1 and Figure A2) revealed that the toxin amount transferred and retained in each tissue was generally higher for STX than for TTX in both the SW and BW groups. In the intestine, TTX amounts (relative value to the dose) averaged 0.2% in the SW group and 0.1% in the BW group at 1 h after administration, and then gradually decreased to below the limit of detection (LOD) at 48 h in the SW group and 24 h in the BW group, while the STX amounts remained high throughout the rearing period, ranging from 5.0 to 8.4% in the SW group and 8.3 to 9.9% in the BW group (Figure 2A). In the liver, TTX was not detected at all throughout the rearing period, whereas a transient transfer was observed for STX, i.e., STX levels increased from 1 to 12 h, peaked at 12 h (1.3% and 2.9% in the SW and BW groups, respectively), and then decreased greatly at 24 h (Figure 2B). In the muscle, STX amounts were generally higher than TTX amounts, with TTX amounts ranging from 0 (below the LOD) to 0.5% in the SW group and 0 to 0.8% in the BW group, while STX amounts ranged from 1.9 to 3.0% in the SW group and 2.1 to 7.5% in the BW group, with a peak at 24 h only in the BW group (Figure 2C). In the skin, TTX levels generally increased from 1 to 24 h, reaching a maximum at 24 h (3.6% and 2.2% in the SW and BW groups, respectively). On the other hand, STX amounts ranged from 2.9 to 4.4% in the SW group and 2.5 to 4.9% in the BW group, and tended to increase throughout the rearing period in the BW group (Figure 2D). In both the SW and BW groups, no abnormal behavior was observed in any of the fish during the rearing period.

The most distinct differences between the two groups were that the BW group showed a faster decrease/disappearance in TTX but greater STX retention in the intestine (Figure 2A), the transient transfer of more STX to the liver (Figure 2B), and a temporary increase in STX as it decayed in the muscle (Figure 2C).

The amount of TTX/STX remaining in the whole body of *C. patoca* is shown in Figure 3. Total TTX amounts ranged from 0.8 to 4.1% in the SW group and 0.8 to 2.4% in the BW group, both with only 2.4% of the administered TTX remaining in the body at the end of the rearing period (48 h). On the other hand, the total STX amounts were 11.7–14.5% in the SW group and 14.9–21.7% in the BW group, reaching 11.8% and 16.2%, 5.0–6.8 times the amount of TTX, respectively, at the end of the rearing period. Both TTX and STX taken up into the body were mainly transferred and accumulated in the skin and muscle, but only STX was also transferred to the liver, and approximately half of the remaining amount in the body was retained in the intestinal tissue.

### 2.2. Micro-Distribution of PSTBPs and Accumulated TTX/STX in the Skin

The PSTBPs distributing to the skin and the TTX/STX accumulating in the skin were visualized by double immunofluorescence labeling with polyclonal anti-PSTBPs and monoclonal anti-TTX or anti-GTX2,3 antibodies (the anti-GTX2,3 antibody also cross-reacts with STX). Strong PSTBP-positive immunofluorescence signals were observed at the epidermal and basal cells in every skin slice of fish from both the SW and BW groups at the end of the rearing period (48 h) (Figure 4). TTX-positive signals overlapped with the PSTBP signal distribution at the epidermal and basal cells with almost the same intensity, whereas STX-positive signals were observed not only at the sites where PSTBPs were distributed, but also at the dermis layer in both groups. In the BW group, however, STX was stained more clearly and extensively than in the SW group, consistent with the results of the toxin analyses of the skin.

## 3. Discussion

Our findings suggest that *C. patoca* has the ability to take up and accumulate both TTXs and STXs, potentially being more selective for STXs than for TTXs, and accumulates STXs more easily, especially under hypoosmotic conditions, which essentially supports our hypotheses. These findings are in contrast with observations in marine pufferfish of the genus *Takifugu*, which selectively take up and accumulate only TTXs, and those in freshwater pufferfish of the genus *Pao*, which selectively take up and accumulate only STXs. Wild *C. patoca* individuals have long been considered to accumulate TTXs as the main toxin. Depending on certain external factors, however, such as an abundance of STXs-rich toxic prey and osmotic pressure in the habitat, *C. patoca* may be able to accumulate larger amounts of STXs over TTXs, even in its natural environment.

Based on oral gavage administration experiments of TTX in pufferfish of the genus *Takifugu*, Wang et al. found that TTX absorbed from the intestine was temporarily retained in the liver and then transferred to the skin [13]. In contrast, in the present study, the amount of TTX in the intestine first decreased and then disappeared in both the SW and BW groups during the 48-h rearing period, whereas the amount of TTX tended to increase in the muscle and skin; interestingly, no TTX was detected in the liver during this period. Mahmud et al. [39] examined the amount of TTX in *C. patoca* collected from Iriomote Island, Okinawa Prefecture, and found high levels of TTX in the skin, muscle, and ovary, but much lower levels in the liver. On the other hand, Tatsuno et al. [14], conducting toxin administration experiments using juvenile (6-month-old) and adult (15-month-old) *T. rubripes*, found that TTX administered in the intestine was transferred and accumulated mainly in the skin in juveniles with undeveloped livers and in the liver in adults with developed livers. While the liver of *C. patoca* is not naturally capable of accumulating abundant TTX, it is possible that little TTX was transferred and retained in the liver in this study because the fish were juveniles.

In contrast to TTX, STX was retained in large amounts in the intestine throughout the rearing period, although transfer to the liver was clearly observed. This transfer was transient, with a sharp decrease in liver STX levels between 12 and 24 h after toxin administration and a corresponding temporary increase in muscle STX levels in the BW group. Fish in the SW group did not show such an increase in muscle STX, but STX tended to increase in the skin in both groups after 12 h, suggesting the unique kinetics of STX from the liver → (muscle) → skin. The reason for the lack of an increase in muscle STX content in the SW group is unclear. The rate of STX transfer may have differed slightly between the SW and BW groups, thus preventing the detection of the peak transfer to muscle in the SW group at the time of sampling due to a gap between the time of sampling and the peak transfer to muscle. In any case, in *C. patoca*, STXs taken up into the body were eventually transferred to the liver and muscle in addition to the skin. Although the gonads were too small for toxin analysis because juvenile fish were used in this experiment, *C. patoca* showed a slightly different uptake and accumulation profile of STX compared with the freshwater pufferfish *P. suvattii*, in which STXs were transferred only to the skin and ovaries in a similar toxin administration experiment [33]. In addition, not only *C. patoca*, but also *P. suvattii* and even *T. pardalis*, a marine pufferfish of the genus *Takifugu*, showed considerable amounts of STXs remaining in the intestine, suggesting that STXs are more easily retained than TTX in the intestinal tissue.

TTX and its analogues 4-*epi*tetrodotoxin (4-*epi*TTX) and 4,9-anhydrotetrodotoxin (4,9-anhydroTTX) are converted to each other, and exist in equilibrium in an approximate ratio of 8:1:1 in an aqueous acidic solution [41]. On the other hand, conversions of STXs components occur in STXs-contaminated bivalves or toxic xanthid crabs [33,42,43]. In this study, however, no TTX derivatives including 4-*epi*TTX and 4,9-anhydroTTX, or no STXs components other than STX were detected in either tissue, and it is unlikely that the bioconversion of TTX/STX takes place in the pufferfish body (this is discussed in detail in our previous paper on the toxin administration experiments [33]). Further studies are needed to evaluate the competitive, inhibitory, and/or synergistic processes during the absorption and distribution of two different toxins (i.e., TTX and STX), as well as their subsequent metabolism in the pufferfish body.

The amount of STX retained in the intestinal tissue, as well as the amount of STX transiently transferred to the liver and finally to the skin was higher in the BW group than the SW group. It is thus presumed that STX uptake from the intestine into the body is enhanced under low-salinity conditions. In general, fish maintain an osmotic pressure of 250–320 mOsm/kg H_2_O (8.3–10.7‰ salinity) in their bodies [44,45]. Thus, marine fish drink seawater to absorb water and excrete excess salt as feces via the intestine, while freshwater fish maintain osmotic balance by obtaining ions from ingested food via the intestine. Under non-iso-osmotic conditions, up to 50% of the energy is used for this osmotic regulation, and under intermediate salinity conditions (8–20‰ salinity), where the osmotic gradient is small and energy costs are low, even individual growth enhancement has been observed [46]. In the BW group, as the salinity of the rearing water (8‰) was closer to the osmotic pressure in the body of *C. patoca* than that in the SW group (33‰), the energy cost of the osmotic adjustment was lower, and it is possible that more small molecules such as STX along with inorganic ions were absorbed from the intestine with the saved energy. On the other hand, TTX decay in the intestine was faster in the BW group than in the SW group. Although the reason for this is not clear, it suggests that the selectivity of TTXs/STXs uptake in the intestine changes (e.g., TTXs selectivity may decrease and STXs selectivity may increase) as the salinity of the environmental water decreases. Sakakura et al. [47] used nontoxic cultured individuals of *T. rubripes* to determine the effect of TTX retention on salinity stress and reported that under very low salinity (1.7‰) conditions, TTX retention could be lethal by adversely affecting osmotic regulation. This is consistent with the fact that freshwater pufferfish, such as those of the genus *Pao*, do not exhibit TTXs uptake or a TTXs accumulation capacity (TTXs selectivity) and, in fact, do not possess TTXs at all. It is possible that these negative TTX characteristics have been an obstacle to the pufferfish’s entry into freshwater, and that the pufferfish evolved to possess STXs instead; euryhaline species such as *C. patoca* and *D. fluviatilis* may be in the process of evolution.

The molecular mechanisms that are involved in the selective uptake and accumulation of TTXs/STXs are still unclear. Yotsu-Yamashita et al. [15] isolated a glycoprotein that binds to STX and TTX, i.e., PSTBP from *T. pardalis* plasma, suggesting its involvement in the absorption, transport, and accumulation of TTX in the pufferfish. PSTBP isoform genes and/or PSTBP-like proteins are widely found in TTX-accumulating species such as *C. patoca* (used in this study) and several pufferfish species of the genus *Arothron* [48], in addition to *Takifugu* marine pufferfish [49,50], but are absent in nontoxic related species of the family Tetraodontidae [50] and freshwater pufferfish of the genus *Pao*, which accumulate only STXs (Yamada et al., unpublished). This observation suggests that PSTBP is not involved in the STX accumulation or selectivity, although PSTBP binds to STX. Tributyltin-binding protein type 2 (TBT-bp2) has a similar structure to PSTBP and is presumed to be the evolutionary origin of PSTBP [50,51]. We recently conducted simultaneous genetic and toxin analyses on freshwater pufferfish of the genus *Pao* from Cambodia, whose species is taxonomically unidentified [26]. We found that different TBT-bp2 sequences at the amino acid level resulted in significantly different STXs accumulation profiles. In addition to this, because marine pufferfish that have STXs, such as *C. patoca* and pufferfish of the genus *Arothron*, also possess diverse TBT-bp2 [48], we speculate that PSTBP is involved in the selective uptake and accumulation of TTX, and TBT-bp2 is involved in the selective uptake and accumulation of STXs, and that the presence/absence and characteristics and/or expression status of the isoform groups lead to the differences in toxin profiles among species and genera.

In double immunofluorescence staining, positive signals for TTX and STX were observed in both epidermal and basal cells, unlike in *T. rubripes*, where TTX-positive signals are detected only in basal cells. In most cases, the distribution of TTX almost overlapped that of PSTBPs, but STX-positive signals were observed in the dermal layer as well as in the areas where PSTBPs were distributed. This suggests that factors other than PSTBPs (i.e., TBT-bp2) are involved in the accumulation of STXs, supporting the previous hypothesis. Further studies are needed to clarify the relationship between TBT-bp2 and the distribution of TTXs/STXs.

## 4. Conclusions

The present study revealed that the euryhaline marine pufferfish *C. patoca* take up and accumulate both TTXs and STXs in their bodies, unlike *Takifugu* marine pufferfish and *Pao* freshwater pufferfish, which selectively take up and accumulate either TTXs or STXs. On the other hand, we found that external factors, including the salinity (osmotic pressure) of the environmental water and the TTXs/STXs abundance in the habitat, influence the toxin profile of pufferfish. Toxin-binding proteins such as PSTBP and TBT-bp2 are implicated in the diversity of toxin selectivity in pufferfish, and therefore, the process by which freshwater pufferfish acquire a different toxin selectivity from marine pufferfish may involve variations in toxin-binding proteins that are linked to adaptations to low osmotic pressure. The exploration of such toxin selectivity expression mechanisms in pufferfish is expected to lead to the identification of toxin accumulation gene groups and studies on the biological system regulation that inhibits the transport and accumulation of toxins in the future. It will further make a significant contribution not only to various academic fields such as fish physiology, toxicology, medicine, pharmacology, and food hygiene, but also to the general public by donating food safety and human health.

## 5. Materials and Methods

### 5.1. Pufferfish Specimens and Acclimation Procedures

Artificially cultured nontoxic individuals of *C. patoca* (body length, 3.15 ± 0.22 cm; body weight, 1.51 ± 0.27 g; *n* = 40) were transported from the Osaka Aquarium NIFREL to Nagasaki University, where they were acclimated for several days in artificial seawater (salinity 33‰) consisting of Marine Art Hi (Tomita Pharmaceutical Ltd., Tokushima, Japan) and dechlorinated tap water. They were then divided into two groups of 20 individuals each (SW and BW groups). The SW group was kept in artificial seawater with 33‰ salinity for five days. On the other hand, for the BW group, the salinity of the rearing water was lowered by 5‰ increments each day from 33‰ to 8‰ over the five days. During these acclimation/rearing periods, Otohime C2 artificial diet (Marubeni Nisshin Feed Ltd., Tokyo, Japan) was fed to the fish twice daily. After one day of acclimation without feeding, the intrarectal TTX/STX administration experiment was conducted.

### 5.2. Toxin Preparation

From the ovaries of *T. rubripes*, TTX with a purity greater than 60% was prepared, and from the xanthid crab *Zosimus aeneus*, STX with a purity greater than 80% was prepared by solvent partitioning, activated charcoal treatment, and Bio-Gel P-2 (Bio-Rad Laboratories, Inc., Hercules, CA, USA) and Bio-Rex 70 (Bio-Rad) column chromatography according to the previously reported method [37], and used for the intrarectal TTX/STX administration experiment. Crystalline TTX (Wako Pure Chemical Industries Ltd., Tokyo, Japan) and an aqueous solution of STX purified from the xanthid crab *Z. aeneus* were used as external standards for the toxin quantification analyses described below.

### 5.3. Intrarectal Toxin Administration Experiment

In our previous TTX/STXs administration experiments using marine/freshwater pufferfish [13,14,33,37], good results were obtained at doses of ~1–2.6 nmol/g body weight. Since the test fish used in this study, however, were very small (body weight, 1.51 ± 0.27 g), we set the dose of both TTX and STX to ~5 nmol/g body weight (7.5 nmol/fish) to ensure the necessary toxin amount for analysis. In a preliminary experiment, when a toxin-containing feed homogenate was administered by oral gavage, the fish spat it out. Therefore, a mixed aqueous solution of TTX and STX (both 375 nmol/mL) was prepared and administered intrarectally to the test fish using a microliter syringe (Hamilton Company, Reno, NV, USA) at a dose of 20 μL/fish (7.5 nmol/fish for both TTX and STX). The SW and BW groups were subsequently reared in 33‰ and 8‰ artificial seawater, respectively, and to observe and compare the relatively short-term toxin kinetics in the body, five individuals from each group were randomly picked up 1, 12, 24, and 48 h after toxin administration to collect intestines, liver, muscle, and skin. Each tissue was homogenized in 0.1% acetic acid, heated in a boiling water bath for 10 min, and centrifuged at 2300 *g* for 15 min. The supernatant was passed through an HLC-DISK membrane filter (0.45 μm, Kanto Chemical Co., Inc., Tokyo, Japan), diluted appropriately to exclude matrix effects, and then used for toxin quantification, as described below. In addition, a portion of the skin of the test fish was collected for double immunofluorescence staining, as described below.

### 5.4. Toxin Quantification

TTX and STX were quantified by liquid chromatography-tandem mass spectrometry and high-performance liquid chromatography with post-column fluorescence derivatization, respectively, according to previously reported methods [37].

For TTX quantification, an Alliance 2690 Separations Module (Waters Corp., Milford, MA, USA) with a Mightysil RP-18 GP column (250 × 2.0 mm, particle size 5 µm, Kanto Chemical Co., Inc.) was used for the chromatography with mobile phase (30 mM heptafluorobutyric acid in 1 mM ammonium acetate buffer, pH 5.0) at a 0.2 mL/min flow rate. The eluate was introduced into a Quattro micro^TM^ API detector (Waters Corp.) in which the TTX was ionized by positive-mode electrospray ionization with a desolvation temperature of 400 °C, a source block temperature of 120 °C, and a cone voltage of 40 V, and monitored at *m/z* 162 (for quantitative analysis) and 302 (for qualitative analysis) as product ions (collision voltage 38 V), with *m/z* 320 as a precursor ion through a MassLynx^TM^ NT operating system (Waters Corp.).

For STX quantification, chromatographic separation was performed using Prominence Ultra-Fast Liquid Chromatography (Shimadzu Corp., Kyoto, Japan) with a LiChroCART^®^ Superspher^®^ 100 RP-18(e) column (250 × 4.0 mm, particle size 4 µm, Kanto Chemical Co., Inc.). STX was separated through the mobile phase containing 2 mM 1-Heptanesulfonic acid sodium salt in 4% acetonitrile-30 mM ammonium phosphate buffer (pH 7.3) at a flow rate of 0.8 mL/min. The column temperature was set at 35 °C. The eluate from the column was mixed continuously with 50 mM periodic acid and 0.2 M KOH containing 1 M ammonium formate and 50% formamide, and heated at 65 °C. Fluorophore formation was monitored at 392 nm with 336-nm excitation through an RF-20A XS Prominence Fluorescence Detector (Shimadzu Corp.).

The LOD and limit of quantification (LOQ) of TTX were 0.0012 nmol/mL (*S/N* = 3) and 0.004 nmol/mL (*S/N* = 10), and those of STX were 0.004 nmol/mL (*S/N* = 3) and 0.013 nmol/mL (*S/N* = 10), respectively.

### 5.5. Immunohistochemistry and Immunofluorescence Staining

For each fish, a small piece of skin was cut out after dissection and then immediately fixed in 10% neutrally buffered formalin. The decalcification, dehydration, and permeation were conducted following the previously reported method [13], and the skin tissues were embedded in paraffin. Afterwards, the skin paraffin sections were cut to a 5-μm thickness using a microtome (HM 335 E, Microm GmbH, Walldorf, Germany). The slides were processed by deparaffinization and hydration, heated for 20 min using antigen unmasking buffer (citrate-based, pH 6.0, 96 °C), and then followed by blocking in 0.01 M phosphate-buffered saline containing 25% goat serum at room temperature. The mouse primary antibody treatment was performed using a monoclonal anti-TTX antibody [52] or a monoclonal antibody against GTX2,3 [53], for which the sensitivity is satisfactory for STX (showing cross-reactivity equivalent to 74% of GTX2,3 [54]) (1 h, room temperature), followed by the incubation with a custom-made rabbit polyclonal antibody raised against a peptide antigen for a PSTBP isoform (designated anti-PSTBPs antibody, Sigma-Aldrich, Tokyo, Japan) for 1 h at room temperature. A VectaFluor^TM^ Duet Double Labeling Kit (DK-8828, Vector Laboratories, Inc., Burlingame, CA, USA) was used as the secondary antibody for 30 min at room temperature, and then the slides were mounted in VECTASHIELD^®^ Antifade Mounting Medium with 4′,6-diamidino-2-phenylindole (DAPI) (H-1800, Vector Laboratories, Inc.). Images of the double immunofluorescence-stained sections were obtained with an all-in-one fluorescence microscope (BZ-X700, Keyence Corp., Osaka, Japan) using a BZ-X700 analyzer software v1.3.0.3 (Keyence Corp.).

## Figures and Tables

**Figure 1 toxins-16-00018-f001:**
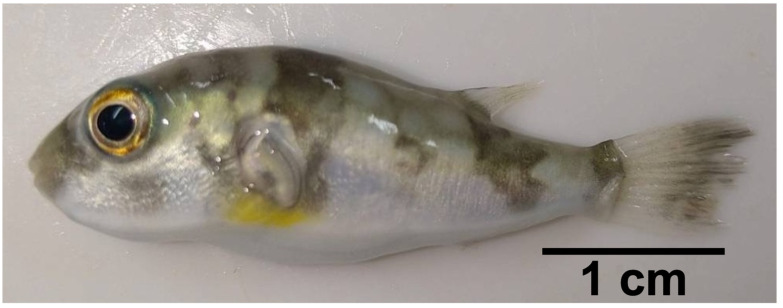
Nontoxic cultured individual of *Chelonodontops patoca*.

**Figure 2 toxins-16-00018-f002:**
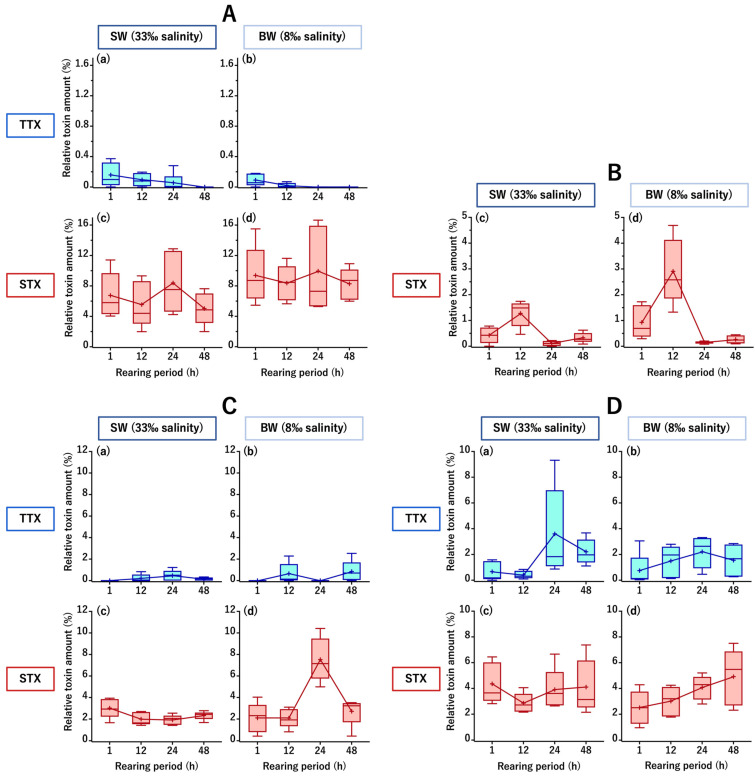
Changes in the relative amounts of tetrodotoxin (TTX) (**a**,**b**) and saxitoxin (STX) (**c**,**d**) (percentage of the administered dose) in the intestine (**A**), liver (**B**), muscle (**C**), and skin (**D**) of *C. patoca* at 1, 12, 24, and 48 h after intrarectal administration of TTX/STX. (**a**,**c**) Represent the seawater group (SW, acclimated/reared at 33‰ salinity) and (**b**,**d**), the brackish water group (BW, acclimated/reared at 8‰ salinity). Data are shown as means (plus signs), medians (inner horizontal lines), upper extremes (bars above), and lower extremes (bars below).

**Figure 3 toxins-16-00018-f003:**
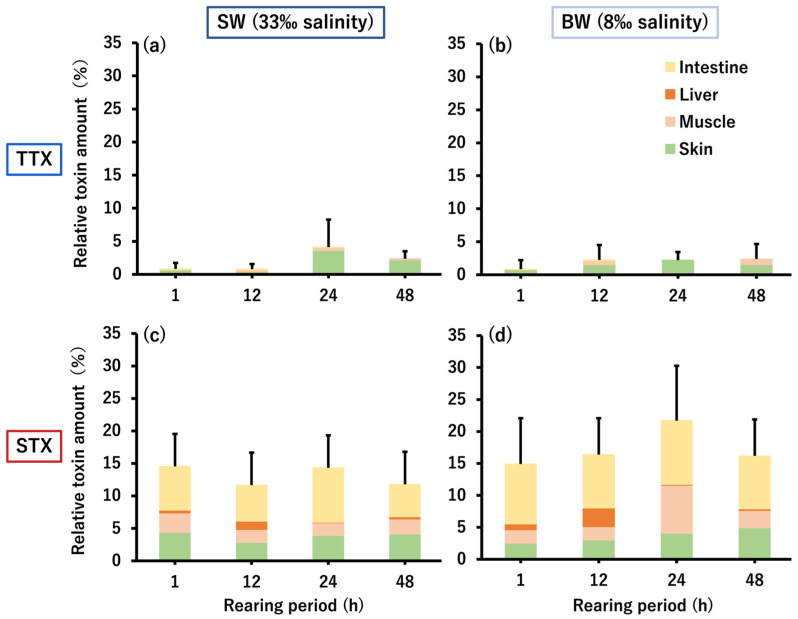
Relative amounts of TTX (**a**,**b**) and STX (**c**,**d**) accumulated in the whole body (intestine, liver, muscle, and skin) of *C. patoca* at 1, 12, 24, and 48 h after intrarectal administration of TTX/STX. (**a**,**c**) Represent the SW group and (**b**,**d**) represent the BW group. Data are shown as means (columns) and standard deviations (SDs) (error bars).

**Figure 4 toxins-16-00018-f004:**
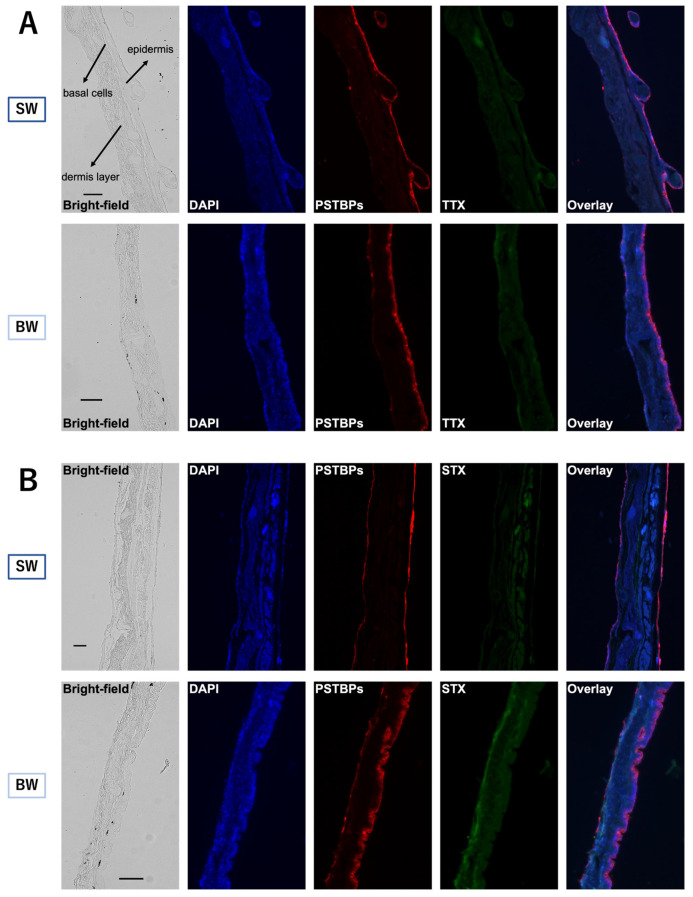
Comparative immunofluorescence staining of skin sections ((**A**) is for TTX and (**B**) is for STX) prepared from an individual from each of the SW (first and third row images) and BW (second and fourth row images) groups of *C. patoca* at 48 h after intrarectal administration of TTX/STX. Blue, red, and green fluorescence represents DNA, pufferfish STX and TTX binding proteins (PSTBPs), and TTX (**A**) or STX (**B**). Scale bars: 100 μm. DAPI: 4′,6-diamidino-2-phenylindole.

## Data Availability

The data presented in this study are available on request from the corresponding author. The data are not publicly available due to privacy or ethical restrictions.

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
