# Peer review of "Tetrodotoxin/Saxitoxin Accumulation Profile in the Euryhaline Marine Pufferfish Chelonodontops patoca"

_toxins, 2023, doi:10.3390/toxins16010018_

Round 1

Reviewer 1 Report

Comments and Suggestions for Authors

Review comments for Tetrodotoxin/saxitoxin accumulation profile in the euryhaline 2 marine pufferfish Chelonodontops patoca

Generally speaking, the paper does a good job explaining analytical data and methods and their explanations appear reasonable.  However, the paper would be improved by the authors making a few changes:

1.  This paper desperately needs a hypothesis.  I have no idea what goal the authors had in doing this work as currently expressed, it reads more like a regulatory research report than the test of a specific idea.

2.  Along those lines, the authors need to justify some of their experimental design with respect to their hypothesis.  In particular I would like to see explanations for why they chose the duration of study they used (seems much shorter than a lot in this literature), an explanation for the dosing level they used (seems a lot lower than most in this literature), and an explanation for the route of toxin administration (most use other types).  I presume these choices are designed to help with testing their particular hypothesis, but since I can't really tell what that is either these things are presented as fait accompli I really have no idea what their thought processes were when making these choices.

It will definitely elevate the paper if they explain their hypothesis and experimental design!

Reviewer 2 Report

Comments and Suggestions for Authors

The observations and comments indicated in the amnuscript must be reviewed and included in the document.

The use of animals in the experiment must comply with animal experimentation directives, licenses, authorizations, etc.

The description of the analytical methods used for the determination of toxins, analytical parameters, are required should be incliude in the document.

The aim of this study must be improved, explaining on a scientific basis the selection of the one toxins dose levels used, and why other level were not used.

If you have data on fish toxicology, behavior, etc., they should be included in this study.

Bibliographic references can be summarized, selecting those most relevant to this study and preferably the most current.

Reviewer 3 Report

Comments and Suggestions for Authors

The manuscript entitled “Tetrodotoxin/saxitoxin accumulation profile in the euryhaline marine pufferfish Chelonodontops patoca” reports the TTX/STX administration experiments using juvenile of the pufferfish C. patoca, in seawater and brackish water conditions. The authors suggested C. pataca accumulates STXs more than TTX, especially under hypoosmotic conditions.

The purpose and result are clear, and the manuscript is well-written. This work is highly worth publishing, as TTX and STX are some of the most important toxins for food safety. Accumulation and kinetics of these toxins in pufferfish would be an issue of interest for broad readers.

This manuscript is suitable for publication in Toxins. However, I would suggest that this manuscript can be improved with several modifications, as below.

1.  I would like to propose adding the representative LCMS and HPLC-FLD chromatograms to the main text or SI. It would underpin the certainty of this work.

2.  Administrated TTX may be converted into chemically equivalent TTX analogs, 4-epiTTX and 4,9-anhydroTTX, during the experiments. If these analogs were detected, please add a note somewhere in this manuscript. Also, I would recommend quantifying the total amount of these TTX analogs if possible. I am curious whether salinity affects the equilibration or not. 

3. It is not mandatory, but I would suggest rearranging the figures. Figures 2, 3, 4, and 5 can be one figure containing four panels because these legends are almost identical. I think it would help the comparison of the results of each tissue. For the same reason, please consider combining Figures 7 and 8. 

Round 2

Reviewer 1 Report

Comments and Suggestions for Authors

No additional comments

Author Response

The authors would like to appreciate your efforts and valuable comments.

Reviewer 2 Report

Comments and Suggestions for Authors

The term PSP or PST is the accepted one, although there are scientists who generally use saxitoxins and is not suitable to rename this group of toxins.

The most appropriate is PST, although PSP can also be used, but recommend use T from toxins.

These natural toxins come from the aquatic environment (marine , freshwaters, etc. ). This natural chemical compounds  reach the first level and lower levels  of the food chain by filtration,  can be transferend along the  feeding  chain and cause the contamination of aquiatic organisms (mussels from freshwaters can be contaminated due to presence of cianobacterial, mussels from sea due to dinoflagelates . shellfish from fresh waters and marine waters can be contaminated due to bloom of cianobacterial or dinoflagelates or others organims.

The term shellfish means different filter-feeding species, such as mussels, freshwater mussels, etc. They are filter-feeding organisms. Therefore, the most appropriate general term for this toxins is shellfish  (PST, Paralytic Shellfish Toxins).

General term of saxitoxins (some paper accept)  give the idea that  include the carbamoylated toxins of STX, NEO, decarbamoylated dcSTX and dcNEO and deoxy (doSTX).

However, other subgroups called  gonyautoxins, not include full term saxitoxin, so if you use term saxitoxins you exclude the concept of gonyautoxins, such as:   C1 to C4 toxins,  GTX1 to GTX6 and dcGTX1 to dcGTX4. This is the reason whay PST acronymous should be recommended and use.

As the work is focused on saxitoxin and tetrodotoxin, the name saxitoxins can be accepted, although it is not appropriate. Scientifically and rigorously, the term should be PST. But the selection should decide the authors and editors.

The same idea ocurring when researchers use terms for Okadaic acid toxins, is incorrect because the expression puit out the concept of dinophysistoxins, etc.

______________________________________________________________________________

APA Style – Requires italics for non-English words, phrases, and abbreviations if they may be unfamiliar to readers, but only on the first use. If the same word, phrase, or abbreviation is used later in the same document, it should be written without italics.

Is et al in italics in Harvard?    

Where a source has one, two or three authors, you should name them all in both your in-text citation and your reference. et al. should always be written in italics, with a full stop at the end of al.   The editors and authors should decid which rule should be aplly. ______________________________________________________________ Please see the notes in TTX chromatograms from LC-MS/MS  and HPLC-FLD for STX, the figures should be improve.
